# The Genetics of Myelodysplastic Syndromes: Clinical Relevance

**DOI:** 10.3390/genes12081144

**Published:** 2021-07-27

**Authors:** Chiara Chiereghin, Erica Travaglino, Matteo Zampini, Elena Saba, Claudia Saitta, Elena Riva, Matteo Bersanelli, Matteo Giovanni Della Porta

**Affiliations:** 1IRCCS Humanitas Research Hospital, Via Alessandro Manzoni 56, 20089 Rozzano, Italy; Chiara.Chiereghin@humanitasresearch.it (C.C.); erica.travagliono@humanitas.it (E.T.); matteo.zampini@humanitasresearch.it (M.Z.); elena.saba@hunimed.eu (E.S.); claudia.saitta@humanitasresearch.it (C.S.); elena.riva@humanitasresearch.it (E.R.); 2Department of Biomedical Sciences, Humanitas University, Via Rita Levi Montalcini 4, 20090 Pieve Emanuele, Italy; matteo.bersanelli@hunimed.eu

**Keywords:** myelodysplastic syndrome, gene mutations, disease classification, prognosis

## Abstract

Myelodysplastic syndromes (MDS) are a clonal disease arising from hematopoietic stem cells, that are characterized by ineffective hematopoiesis (leading to peripheral blood cytopenia) and by an increased risk of evolution into acute myeloid leukemia. MDS are driven by a complex combination of genetic mutations that results in heterogeneous clinical phenotype and outcome. Genetic studies have enabled the identification of a set of recurrently mutated genes which are central to the pathogenesis of MDS and can be organized into a limited number of cellular pathways, including RNA splicing (*SF3B1*, *SRSF2*, *ZRSR2*, *U2AF1* genes), DNA methylation (*TET2*, *DNMT3A*, *IDH1/2*), transcription regulation (*RUNX1*), signal transduction (*CBL*, *RAS*), DNA repair (*TP53*), chromatin modification (*ASXL1*, *EZH2*), and cohesin complex (*STAG2*). Few genes are consistently mutated in >10% of patients, whereas a long tail of 40–50 genes are mutated in <5% of cases. At diagnosis, the majority of MDS patients have 2–4 driver mutations and hundreds of background mutations. Reliable genotype/phenotype relationships were described in MDS: *SF3B1* mutations are associated with the presence of ring sideroblasts and more recent studies indicate that other splicing mutations (*SRSF2*, *U2AF1*) may identify distinct disease categories with specific hematological features. Moreover, gene mutations have been shown to influence the probability of survival and risk of disease progression and mutational status may add significant information to currently available prognostic tools. For instance, *SF3B1* mutations are predictors of favourable prognosis, while driver mutations of other genes (such as *ASXL1*, *SRSF2*, *RUNX1*, *TP53*) are associated with a reduced probability of survival and increased risk of disease progression. In this article, we review the most recent advances in our understanding of the genetic basis of myelodysplastic syndromes and discuss its clinical relevance.

## 1. Introduction

In the World Health Organization (WHO) classification of myeloid neoplasms, myelodysplastic syndromes (MDS) are defined as clonal disorders of hematopoietic stem cell progenitors characterized by morphologic dysplasia, ineffective hematopoiesis (leading to peripheral blood cytopenia), and increased risk of evolution into acute myeloid leukemia [1].

Myelodysplastic syndromes (MDS) typically occur in elderly people. The natural history of MDS is heterogeneous ranging from conditions with a near-normal life expectancy to forming rapidly evolving into acute myeloid leukemia. In such a heterogeneous disease, a risk-adapted treatment strategy is mandatory [2,3].

Currently, several prognostic systems can be used to assess disease risk, including the International Prognostic Scoring System (IPSS), and its revised version (revised IPSS, IPSS-R), that are mainly based on clinical and hematological parameters (i.e., severity of peripheral blood cytopenias, percentage of bone marrow blasts, and presence of cytogenetic abnormalities) [4,5]. These prognostic tools present limitations, and in some cases fail to capture reliable prognostic information at the individual patient level [6].

Several therapeutic options have been proposed for MDS patients but only few treatments survived the evidence-based criteria of efficacy. Erythropoiesis stimulating agents (ESA) are useful for improving anemia in the early disease stages. Allogeneic transplantation (HSCT) is the only potentially curative treatment for MDS patients; however, due to a not negligible morbidity and mortality associated with the procedure, an accurate selection of candidate patients is required. Hypomethylating agents (HMA) are approved for the treatment of high risk MDS and may improve survival in patients not eligible to transplantation [7].

The growth and spread of a somatically mutated clone represent the pathophysiological process that leads to MDS. The selective advantage of the clone is provided by acquired (somatic) genetic lesions (driver mutations) [8,9,10]. Several driver mutations, belonging to different cellular pathways, can induce a MDS phenotype, and the great majority of patients have a complex combination of different gene mutations, accounting for the clinical heterogeneity of the disease [11,12,13,14,15,16]. Increasing scientific evidence suggests that mutational screening may improve disease classification and prognostication, thus enabling the refinement of clinical decision making in these disorders [14,15,16,17].

Over the past decade, the development of new cost and time-effective Next-Generation-Sequencing (NGS) techniques have led to a new genomic-era in cancer research where mutational profiling has entered in the clinical practice as part of the decision-making of patients with MDS [8,9,10]. In this review, we discuss the emerging role of mutational screening in the diagnosis, prognostication, and treatment of MDS patients.

## 2. Clonal Hematopoiesis of Indeterminate Potential (CHIP)

Somatic mutations can occur in hematopoietic stem cells at a low frequency during their life. Recent findings suggest that mutational processes are largely independent of cell division and are important contributors to somatic mutagenesis associated with aging process [18]. Any genetic alteration that causes a selective advantage relative to other self-renewing cells will lead to a clonal dominance. These mutated hematopoietic stem cells may acquire additional genetic lesions, which induce an overt hematological phenotype (such as MDS or other myeloid neoplasms). These consequences are amplified in the elderly because the aging process itself may not only deplete hematopoietic stem cells, but also alter the marrow microenvironment [19].

Skewing of X-chromosome inactivation has been found in a significant proportion (40%) of healthy women aged >60 years. A subset of these women was found to carry mutations in the *TET2* gene, suggestive of clonal hematopoiesis driven by a somatic mutation [20]. In more recent large studies, exome sequencing of peripheral blood from thousands of subjects without hematologic malignancies has identified the age-dependent clonal expansion of somatic mutations in the hematopoietic system that was associated with an increased risk of cancer and other diseases [21,22,23,24,25]. This phenomenon has been termed “clonal hematopoiesis of indeterminate potential” (CHIP) [26].

Most frequent mutations related to clonal hematopoiesis are reported in three chromatin-related genes: *DNMT3A*, *TET2*, and *ASXL1*. Additional recurrent molecular abnormalities associated to CHIP include mutations in genes encoding for RNA splicing factors, which are also frequently reported in patients affected with MDS and other myeloid neoplasms [21,22,23,24,25,26]. Interestingly, the mutation frequencies are age-dependent— mutations in any of these genes are found in ≤1% of people aged <50 years but in ≥10–20% of people aged >65 years. A strong association has been found between somatic mutations and the future development of cancers: where they were present, there was a significantly higher risk (10-fold) for subsequent hematological malignancies with respect to individuals without clonal hematopoiesis. Somatic variants also increased the risks of inflammation-related chronic disease (such as coronary heart disease and stroke) and death [21,22,23,24,25,26].

The time and place of individual mutations and their clonal emergence during the course of the disease are central issues for a better comprehension of MDS pathogenesis, for the development of cancer preventive strategies, and for the design of potentially new therapies to eradicate clones harbouring the genetic aberrations that accumulate in hematopoietic stem cell progenitors [10]. In this context, treatment with vitamin C induces *TET2* restoration in *TET2*-deficient mouse hematopoietic progenitors and is able to suppress leukemia progression [27]. Moreover, H3B-8800, an orally available splicing modulator molecule is able to induce lethality in spliceosome-mutant cells [28].

At the moment there are no data enough to suggest CHIP screening in asymptomatic patients in clinical practice. In fact, the presence of mutations “per se” in a given individual has only limited predictive power as conversion to overt diseases is rare regardless of mutational status [8,9,10,26]. Moreover, additional non-mutational factors may be responsible for the induction of an MDS. In a model of clonal evolution starting with CHIP and ending in an overt hematologic malignancy, the transition to MDS involves a complex interaction between epigenetic alterations within the hematopoietic stem cell and a dysfunctional bone marrow microenvironment [8,9,10,26].

## 3. Recurrently Mutated Genes in Myelodysplastic Syndromes

In the last years the advent of methods that improved genome/exome sequencing costs and throughputs allowed a detailed knowledge of the different mutational landscapes in MDS patients [8,9,10]. Using targeted-sequencing approaches, large MDS cohorts have been characterized by their mutational profiles and several recurrently mutated genes have been associated to myeloid neoplasms. By using this approach, up to 90% of patients have been found to have a somatic mutation in at least one gene, while the great majority of patients carried 2–4 mutations. Only a few genes are consistently mutated in >10% MDS patients, whereas a long tail of 40–50 genes are mutated less frequently (<5% of cases) [14,15,16]. The number of mutated genes in MDS are pretty high, but they are implicated in few biological pathways including: RNA splicing factors, epigenetic regulators, signal transduction, transcription factors, DNA damage response, and cohesin components [14,15,16].

### 3.1. RNA Splicing Mutations

Spliceosome components (*SF3B1*, *SRSF2*, *U2AF1*, and *ZRSR2* genes) are mutated in 50–60% of patients affected with MDS [12,13]. Spliceosome mutations are rarely observed in childhood myeloid neoplasms, suggesting that they are specifically acquired in elderly people [10]. Mutations of spliceosome are founding genetic lesions and are mutually exclusive to each other. In fact, the mutant allele burden is usually 40–50%, indicating a dominant clone in the bone marrow that is heterozygous for the mutation [12,13]. Rarely, MDS cases with >1 splicing factor mutation have been reported, highlighting allele-specific differences as a critical factor in regulating the molecular effects of RNA splicing mutations as well as their co-occurrences/exclusivities with one another gene [29]. *SF3B1*, *SRSF2*, and *U2AF1* genes are mainly characterized by missense mutations in few mutational hotspots, while non-sense or frameshift changes have not been described [15,16]. Different spliceosome mutations are associated with specific clinical phenotypes and probability of overall survival/risk of leukemic evolution. Somatic *SF3B1* mutations are strong associated to MDS patients with ring sideroblasts with/without thrombocytosis suggesting, a causal relationship between *SF3B1* mutation and formation of ring sideroblasts [15,16,30]. In addition, the great majority of *SF3B1* mutated MDS patients showed a favourable clinical outcome and low risk of leukemic transformation [15,16,30]. *SRSF2* mutations are mainly associated to MDS characterized by multilineage dysplasia in the bone marrow and/or excess blasts and predict poor prognosis and a high risk of leukemic evolution [15,16]. Somatic mutations of *U2AF1* have been described in different MDS subtypes (mainly including forms with multilineage dysplasia and excess of blasts) and are predictive of a high risk of leukemic evolution and poor survival [15,16].

### 3.2. Epigenetic Regulators

Mutations in genes involved in the epigenetic regulation of transcription are very common in patients affected with MDS. In particular, in DNA methylation associated genes such as the de novo DNA methyltransferase DNMT3A and the methylcytosine dioxygenase TET2, recurrent missense, nonsense, splice site, and frameshift mutations have been identified [31,32,33]. Loss-of-function mutations of components of histone modification complexes (*ASXL1 EZH2*) are reported in 20% and 5% of patients, respectively [34,35]. *ASXL1* mutations are common in several myeloid neoplasms as well as MDS and are associated with poor outcome [34].

### 3.3. Mutations in Other Cellular Pathways

Somatic mutations of transcription factors have been observed in MDS patients. The *RUNX1* gene is mutated in 7–10% of patients and it is associated with advanced disease, moderate-to-severe thrombocytopenia, and poor clinical outcome [14,15,16]. Somatic mutations in the tumor suppressor gene *TP53* gene, mapping on chromosome 17p13.1, have been identified in different types of cancer [35]. *TP53* mutations are found in 5–10% of MDS patients affected and are associated with an excess of blasts and complex karyotype (including abnormalities of chromosome 17 or deletions of chromosome 5 and 7) [14,15,16]. *TP53*-mutated MDS patients have an unfavourable clinical outcome and a high risk of disease progression, and the same is true for patients with other myeloid neoplasms carrying *TP53* mutations [36]. The paradigmatic example of clinical relevance of *TP53* mutations in MDS is provided by MDS with del(5q): subclones carrying *TP53*-mutations may occur at an early disease stage in MDS with del(5q) and are closely associated with a poor response to lenalidomide and with an increased risk of leukemic evolution [37].

Cohesin is a multi-subunit protein complex involved in the 3D shaping of the human genome and plays a critical role in the regulation of transcription and in several DNA repair mechanisms. The 4–5% of MDS patients have been found mutated in a gene coding for one of its subunits (*STAG2*). In patients affected with acute myeloid leukemia, a comparable frequency of *STAG2* mutation rate has been reported, suggesting that altered cohesin function may have a role in myeloid leukemogenesis process [38].

## 4. Molecular Classification of Myelodysplastic Syndromes

The mutational landscape in MDS patients at diagnosis is characterized by the presence of 2–4 oncogenic driver mutations and hundreds of background or passenger mutations. Studies on variant allele frequency identified the association of mutations in RNA splicing and DNA methylation genes in the early phase of clonal proliferation, whereas other gene categories are mainly involved in the subsequent clonal evolution. However, the temporal order of acquisition of different driver mutations is not fixed and may significantly vary from subject to subject [14,15,16].

Some specific genotype/phenotype correlations have been described in MDS. The most relevant finding in this context is the close relationship between *SF3B1* mutations with MDS with ring sideroblasts, which may provide the rationale for a molecular classification of these disorders [39].

Current disease classification provided by World Health Organization (WHO) mainly uses morphological features to define MDS subtypes, leading to a clinical overlap between different categories. Moreover, a low inter-observer reproducibility in the morphological evaluation of bone marrow dysplasia is observed in clinical practice [1,40]. In myeloid neoplasms, classifications based on clinical and morphologic criteria are being complemented by the introduction of specific genomic features, which may capture better clinical-pathological entities [1].

Recently, a first example of molecular classification of MDS was proposed, on the basis of a retrospective study of 2043 patients [41]. In the study, both gene mutations and cytogenetic abnormalities were combined, identifying eight MDS subgroups that shared specific genomic and clinical features. In five subgroups, splicing gene mutations (*SF3B1*, *SRSF2* and *U2AF1*) were identified as the dominant genomic features. Mutations in these genes occur early in disease history and they determine specific clinical phenotypes driving different disease evolution patterns. MDS categories defined by splicing gene mutations display different prognosis (groups with *SF3B1* mutations being associated with better probability of survival) [41] (Table 1).

More in details, *SF3B1* mutations define a specific MDS subgroup characterized by ring sideroblasts, a low percentage of bone marrow blasts and with favourable outcome. Among *SF3B1*-mutated patients, the acquisition of a myeloproliferative phenotype (often characterized by progressive thrombocytosis) has been induced by the simultaneous presence of *JAK/STAT* pathway mutations. Another subgroup includes patients carrying *SF3B1* mutations with co-existing mutations in other genes (more frequently *RUNX1* and *ASXL1*), and it is usually characterized by multilineage dysplasia, higher bone marrow blast count, and poorer outcome. *SRSF2* and *U2AF1* mutations characterize distinct disease categories with specific co-mutation patterns, clinical phenotype, and with reduced survival with respect to *SF3B1*-defined categories [41].

The subgroup of MDS associated with *TP53* mutations and/or complex karyotype is characterized by a very poor prognosis [36,41].

A further MDS category includes patients with mutations that are recurrently described in de novo acute myeloid leukemias (*NPM1*, *FLT3*, *IDH1*, and *RUNX1*); this category is associated with a high risk of disease progression and poor outcome, suggesting that the current threshold of 20% marrow blasts included in the current WHO classification of myeloid neoplasms might be not appropriate to recognize different biological distinct disease categories [41]. Moreover, the subgroup without specific genomic features includes a high percentage of MDS with bone marrow hypocellularity that share features with aplastic anemia. Overall, these findings indicate that a genomic classification could transcend the boundaries of MDS classification and could shed light on those cases overlapping with other myeloid conditions where current morphological criteria are often inadequate [41].

## 5. From Molecular Classification to Next-Generation Prognostic Scores in Myelodysplastic Syndromes

MDS is a very heterogeneous disease with a very different risk of leukemic transformation and survival. Several prognostic risk stratification systems have been developed to facilitate clinical decision-making [2,3,7]. The most widely used tools are the International Prognostic Scoring System (IPSS), and its revised version (revised IPSS, IPSS-R). Both scores are based on bone marrow morphology to provide bone marrow blast count, conventional cytogenetics to detect clonal chromosomal abnormalities, and the degree of cytopenias to stratify disease-related risk [4,5]. However, the prognostic subgroups identified by these tools are not able to optimally resolve patient outcomes variability [6].

To date, the prognostic effect of somatic mutations in MDS patients has been examined in several large and cooperative studies. Although these studies differed by statistical methods used, the number of sequenced genes, size and composition of patient cohorts, some common features have been identified [12,13,14,41].

The clinical outcome is strongly influenced by the gene of the first founder mutations of the initial clone. As an example, MDS with ring sideroblasts may be originated by founding mutations both in *SF3B1* gene and in *SRSF2* gene, but the median probability of survival is 10 years in the former vs. <2 years in the latter. Moreover, the early diagnosis of MDS progression and/or leukemic evolution can be performed through the detection of subclonal mutations. Finally, in the MDS prognosis, not only the type of genes mutated but also the number of somatic variants occurred per patient is important—the higher the number of mutated genes, the poorer the prognosis [8,12,13,14,41].

Across different studies, *TP53*, *RUNX1*, *ASXL1*, *EZH2*, *SRSF2*, and *ETV6* mutations are associated with poor probability of survival, whereas *SF3B1* mutations predict better clinical outcomes and low risk of disease evolution. Interestingly, somatic mutations can estimate survival independently of clinical prognostic scoring systems (i.e., IPSS and IPSS-R). However, given that morphology, bone marrow blast count, and peripheral blood cytopenias are likely closely linked to the genotype of the MDS clone, it follows that those prognostic systems that consider a detailed set of clinical and hematological features are only few, improved by the inclusion of mutational screening [8,12,13,14,41]. Therefore, we can expect that morphologic and clinical criteria will continue to have a central role in defining individual MDS prognosis [41]. In light of this, before the routine implementation of mutational screening for MDS prognostication, further investigations are warranted. Multicentric studies, collecting comprehensive clinical annotation and genomic features in large MDS patient populations are required to correctly integrate genetic information into existing prognostic systems [8,9,10].

A basic statistical approach may not be effective to develop robust decision support systems, and therefore, innovative higher-level statistic methods are required to control for confounding factors and analyze the many variables that may have clinical significance [41,42,43,44]. In this context, new models were developed to produce individual tailored survival prediction scores using both clinical and genomic information in myeloid neoplasms including MDS [41,42,43,44]. While conventional prognostic tools provide an outcome prediction calculated through the median probability of the survival of the patient groups that share similar clinical features, these new prognostic systems use individual patient genotype and phenotype. These approaches allow for a personalized prediction of clinical outcome in order to significantly improve the capability of acquiring prognostic information in such a heterogeneous disease [41,42,43,44].

Identification of somatic variants is not central only in MDS pathogenesis, but it could also be useful in the treatment decision-making process. An illustrative example of the use of genetics in this step has been illustrated in MDS carrying del(5q). These patients are treated with lenalidomide that usually lead to a cytogenetic complete remission and a significant improvement of anemia [7]. Del(5q) MDS patients with TP53 mutations have shown a reduction in lenalidomide response and a simultaneous expansion of *TP53*-mutant subclones [37].

To date, the only potentially curative therapy for MDS is allogeneic stem cell transplantation [7]. Accurate pre-transplant risk assessment is needed to estimate the success rate for this procedure in order to prevent unnecessary morbidity/mortality in those patients that are unlikely to benefit [7]. Although clinical factors such as bone marrow blast count and cytogenetics have been shown to influence post-transplantation outcome, there is also an emerging role for genetics in this regard [8,9,10]. In three large retrospective studies of patients who underwent pre-transplant genetic profiling, mutations in *TP53* were associated with significantly decreased overall survival [45,46,47]. In younger MDS patients, inherited mutations in *SBDS* gene were unexpectedly common (4%) and showed a close connection with somatic *TP53* mutations, indicating a biologic synergy between *SBDS* and *TP53* gene lesions in the clonal MDS transformation of Shwachman–Diamond syndrome [46]. More recently, it was shown that genomic features (including cytogenetics, gene, and gene-gene interactions) are relevant for predicting survival after transplantation, improving the level of scientific evidence to add this information in MDS transplantation decision making [41].

Although new validations are needed in large clinical cohorts, genetics data will be central in the identification of the particular subsets of patients in whom standard transplant regimens are particularly ineffective and in whom alternative therapeutic strategies should be considered.

Moreover, thanks to the recent studies that linked somatic mutations to the patophy-siology of MDS, new molecular therapeutic agents have been developed. For instance, in recent clinical studies, isocitrate dehydrogenase enzymes (IDH) mutant MDS patients have benefited from *IDH*-mutation small-molecule inhibitors treatment [33]. Moreover, thanks to animal models of splicing factor mutations, some studies have discovered that inhibition of the splicing machinery can drive cell death [39]. In this context, new spliceosome inhibitors are in development to strengthen the therapeutic approaches to MDS patients carrying these mutations [39].

## 6. Conclusions

While scientists are accumulating evidences to better understand the biological links among somatic mutations, diagnosis, prognosis, and therapeutic approaches in MDS patients, physicians are trying to better integrate genomics data into clinical practice.

Genes identified as being recurrently mutated in large myeloid neoplasms cohorts have been used to construct new targeted sequencing panels. Moreover, new scenarios are emerging where advances obtained in recent studies have been used to improve the clinical practice.

Some practical example, of molecular classification of MDS are ongoing, providing evidence of specific relationship between distinct mutational patterns, phenotype, and disease evolution. For example, *SF3B1* mutations define a subset of MDS patients with ring sideroblasts and favourable prognosis, and now this specific mutation is included formally into diagnostic criteria of WHO classification. More recent scientific evidences suggest that splicing mutations may define additional disease categories with specific clinical and hematological features. Mutations in *TP53* define a distinct MDS entity, associated with adverse outcomes and with poor response to conventional treatments, including transplantation. Moving forward, large, prospective studies will enable progress toward the goal of effective personalized treatment strategies based on disease genotype.

## Figures and Tables

**Table 1 genes-12-01144-t001:** Molecular classification of myelodysplastic syndromes. MDS del(5q), MDS with isolated deletion of long arm of chromosome 5; MDS-SLD, MDS with single lineage dysplasia; MDS-MLD, MDS with multilineage dysplasia; MDS-RS-SLD, MDS with ring sideroblasts and single lineage dysplasia; MDS-RS-MLD, MDS with ring sideroblasts and multilineage dysplasia; MDS-EB1, MDS with excess of blasts, type 1; MDS-EB2, MDS with excess of blasts, type 2.

		Genomic-Based MDS Category	Clinical and Hematological Features	WHO 2016 MDS Categories	Prognosis
MDS associated with splicing gene mutations	*SF3B1*-related MDS	MDS with isolated *SF3B1* mutations (or associated with mutations of clonal hematopoiesis and/or *JAK/STAT* pathways genes)	- Peripheral blood: isolated anemia, normal to high platelet count - Bone marrow: single or multilineage dysplasia, ring sideroblasts, low percentage of bone marrow blasts	MDS-RS-SLD; MDS-RS-MLD	Very good prognosis
		MDS with *SF3B1* and co existing mutations (including *RUNX1*, *ASXL1*)	- Peripheral blood: anemia, mild neutropenia, thrombocytopenia - Bone marrow: multilineage dysplasia, ring sideroblasts, excess blasts	MDS-RS-MLD, MDS-EB1, MDS-EB2	Good prognosis (less favourable as compared to MDS with isolated *SF3B1*)
	*SRSF2*-related MDS	MDS with *SRSF2* and concomitant *TET2* mutations	- Peripheral blood: single cytopenia (anemia in most cases), higher monocyte absolute count - Bone marrow: multilineage dysplasia, excess blasts	MDS-MLD, MDS-EB1, MDS-EB2	Worse prognosis with respect to *SF3B1*-related groups
		MDS with *SRSF2* mutations and co-existing mutations in other genes (*ASXL1*, *RUNX1*, *IDH2*, and *EZH2*)	- Peripheral blood: two or more cytopenias - Bone marrow: multilineage dysplasia, excess blasts	MDS-EB2	Poor prognosis (Worse prognosis with respect to MDS with *SRSF2* and *TET2* mutations)
	*U2AF1*-related MDS	MDS with *U2AF1* mutations associated with deletion of chromosome 20q, and/or abnormalities of chromosome 7	- Peripheral blood: severe transfusion-dependent anemia - Bone marrow: multilineage dysplasia, excess blasts	MDS-MLD, MDS-EB1, MDS-EB2	Poor prognosis
		MDS with *TP53* mutations and/or complex karyotype	- Peripheral blood: two or more cytopenias with transfusion-dependency - Bone marrow: excess blasts	MDS-EB1, MDS-EB2	Very poor prognosis, high rate of leukemic evolution
		MDS with AML-like mutations (*DNMT3A*, *NPM1*, *IDH1*, *RUNX1*)	- Peripheral blood: two or more cytopenias with transfusion dependency - Bone marrow: excess blasts	MDS-EB1, MDS-EB2	Poor prognosis, high rate of leukemic evolution
		MDS without specific genomic profiles	- Peripheral blood: asympotmatic anemia - Bone marrow: normal to reduced bone marrow cellularity, no ring sideroblasts, low percentage of marrow blasts	MDS-SLD; MDS-MLD	Good prognosis
	MDS del(5q)	MDS with isolated 5q, with none or one mutation (excluding *TP53*)	- Peripheral blood: mild anemia without transfusion dependency -Bone marrow: multilineage dysplasia, low percentage of bone marrow blasts	MDS del(5q)	Good prognosis
		MDS with isolated 5q with two or more mutations or *TP53* mutations	- Peripheral blood: mild anemia - Bone marrow: multilineage dysplasia, no excess blast	MDS del(5q)	Worse prognosis and higher rate of leukemic evolution with respect to MDS del(5q) with none or one mutation

## Data Availability

Not Applicable.

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
