# Peer review of "The Genetics of Myelodysplastic Syndromes: Clinical Relevance"

_genes, 2021, doi:10.3390/genes12081144_

Round 1

Reviewer 1 Report

This is a nicely written and comprehensive review article describing the most recent advances in the understanding of the genetic basis of the myelodysplastic syndromes and the clinical relevance of the recurrently mutated genes identified in these disorders.

The authors mention that mutations of the RNA splicing machinery are largely mutually exclusive and are most often founding events (lines 131-132). Here, I would suggest adding that rare MDS cases with more than one splicing factor mutation have been reported in some studies. The study by Taylor et al. (Blood 2020 - PMID: 32640014) showing that mutual exclusivity and co-occurrence of splicing factor mutations is allele-specific rather than gene-specific should then be mentioned and referenced.

Other minor comments

- Gene names should be italicized throughout the manuscript

- Line 104: please correct “fond” to “found”

- Line 179: please correct “in;6%” to “in 6%”

- Line 267: please correct “complimented” to “complemented”

Reviewer 2 Report

This is a well written and informative review on the clinical relevance of somatic mutations in MDS. The manuscript incorporates key recent literature in the field. I only have a few minor comments.

  1. Line 71: Mutational processes that are independent of cell division are also clearly important contributors to somatic mutagenesis (see Abascal et al., Nature volume 593, pages 405–410 (2021)). Please reference this important recently published work.

  1. Line 74: What is the evidence to support the statement-“The consequence of this abnormality is genomic instability leading to increased risk of acquiring additional mutations inducing overt MDS phenotype”? It may be true in the context of acquired biallelic TP53 mutations (see Bernard et al 2020) but the authors should justify the statement with other relevant supporting references.

  1. It is important to highlight to the reader the importance of excluding germline mutations in younger patients presenting with MDS (eg Lindsley NEJM 2017; Rio-Machin et al. Nature Communications volume 11, 1044 (2020)). They are an important subset of those MDS patients who have reduced survival due to the presence of concurrent somatic TP53 mutations.
